# Improving the understanding of cytoneme-mediated morphogen gradients by *in silico* modeling

**Adrián Aguirre-Tamaral**[ID]*, **Isabel Guerrero**[ID]*

Tissue and Organ Homeostasis, Centro de Biología Molecular "Severo Ochoa" (CSIC-UAM), Universidad Autónoma de Madrid, Cantoblanco, Madrid, Spain

* adrian.aguirre@cbm.csic.es (AA-T); iguerrero@cbm.csic.es (IG)

**Data Availability Statement:** The software code and a manual for users are available in the software repository: https://github.com/AdrianA-T/cytomorph.

## Abstract

Morphogen gradients are crucial for the development of organisms. The biochemical properties of many morphogens prevent their extracellular free diffusion, indicating the need of an active mechanism for transport. The involvement of filopodial structures (cytonemes) has been proposed for morphogen signaling. Here, we describe an *in silico* model based on the main general features of cytoneme-mediated gradient formation and its implementation into Cytomorph, an open software tool. We have tested the spatial and temporal adaptability of our model quantifying Hedgehog (Hh) gradient formation in two *Drosophila* tissues. Cytomorph is able to reproduce the gradient and explain the different scaling between the two epithelia. After experimental validation, we studied the predicted impact of a range of features such as length, size, density, dynamics and contact behavior of cytonemes on Hh morphogen distribution. Our results illustrate Cytomorph as an adaptive tool to test different morphogen gradients and to generate hypotheses that are difficult to study experimentally.

## Author summary

Graded distribution of signaling molecules (morphogens) is crucial for the development of organisms. Signaling membrane protrusions, called Cytonemes, have been experimentally demonstrated to be involved in morphogen transport and reception. Here, we have developed an *in silico* model for gradient formation based on key features of cytoneme mediated signaling. We have also implemented the model into an open software tool we named Cytomorph, and validated it by comparing its simulations with experimental data obtained from Hedgehog morphogen distribution. Finally, we have generated *in silico* predictions for the impact of different cytoneme features such as length, size, density, dynamics and contact behavior. Our results show that Cytomorph is an adaptive tool that can facilitate the study of other cytoneme-dependent morphogen gradients, besides being able to generate hypotheses about aspects that remain elusive to experimental approaches.

**Funding:** This work was funded by the Ministerio de Economía, Industria y Competitividad, Gobierno de España (MINECO) to I.G, BFU2014-59438-P, by the Ministerio de Economía, Industria y Competitividad, Gobierno de España (MINECO) to A.A-T, (FPI) BFU2014-59438-P, the Ministerio de Ciencia, Innovación y Universidades to I.G, BFU2017-83789-P, and the Ministerio de Ciencia, Innovación y Universidades to I.G, RED2018-102411-T. The funders had no role in study design, data collection and analysis, decision to publish, or preparation of the manuscript.

**Competing interests:** The authors have declared that no competing interests exist.

## Introduction

During embryonic development, groups of cells are organized to give rise to tissues and organs. Precise spatio-temporal control of cell-to-cell communication is needed during proliferation, cellular three-dimensional organization and differentiation. Misregulation of these events is one of the most prevalent causes of diseases such as congenital malformations, neurological disorders and cancer [1]. Several signaling molecules are defined as morphogens, messengers that are transported at a distance in a concentration-dependent manner to regulate the differential activation of target genes [2]. The cellular mechanisms involved in the transport of the morphogens are still under debate [3].

Modeling has been a useful strategy to explore complex biological processes. Models to explain pattern generation during development have been mainly focused on the description of how, when and where a morphogenetic signal induces a specific cellular response within a particular tissue. Especially relevant were the early works of Turing [4] and Wolpert [5], who set the foundations of how a precise morphogen distribution could determine cell fate and patterning in a concentration-dependent manner. Subsequently, several works took into account the effect of the production and degradation of morphogens [6–9] and their transport was usually modeled by inferring a diffusion mechanism [10]. Since the molecular properties of most morphogens impede them to diffuse freely in the extracellular environment, a different mechanism for their transport is required [11]. A transport mechanism based on cytonemes (filopodia-like-structures) has been observed for most signaling pathways [12–15]: Decapentaplegic (Dpp) [16,17], Wingless (Wnt/Wg) [18,19], Epithermal Growth Factor (EGF) [20], Fibroblast Growth Factor (FGF) [20], Hedgehog (Hh) [21–23] and Notch [24–27]. Cytonemes are actin-based membrane protrusions emanating from morphogen producing and/or receiving cells that deliver and/or collect morphogen by direct cell-cell membrane contacts (Fig 1). Increasing experimental evidences by live imaging in developing tissues highlight the implication of dynamic cytonemes in short- and long-distance cell communication [22,23,28–30].

A few mathematical models centered on different aspects of cytoneme-mediated signaling have been proposed (reviewed in [31]). They focus in characteristics such as vesicular transport along cytonemes [32,33], cytoneme contact mechanisms [34] or cytoneme guidance towards correct target receiving cells in a 1D system [35]. To date there are also some models concerning the cytoneme-mediated establishment of morphogen gradient during pattern formation [36–38]. Those models use static cytonemes and weight functions pondering the quantity of morphogen received. However, experimental evidence indicates that cytoneme dynamics can play an important role [22,23,28–30] and at present there are no data sustaining a pondered mechanism of signaling. Those models also assume a local source of morphogen, which is not true in most cases and theoretical studies emphasize the importance of using an extended source [39]. Finally, previous models are not computationally implemented into a tool that can be used to test or load experimental data on cytoneme-mediated morphogen gradients.

In this work, we have developed a new dynamic model for cytoneme-mediated gradient formation in compartmentalized tissues during development, which validates this mechanism of cell signaling and has several advantages: 1) it was designed to be general enough to be applied to different morphogens or tissues, 2) it considers an extended morphogen source within a developing tissue, 3) signaling is not based on weighted mechanisms, 4) it considers the dynamics of cytonemes and 5) it has been implemented into a computational tool (Cytomorph), which facilitates the development of *in silico* predictions for different morphogens.

Finally, we tested and validated our computational model for the Hh signaling pathway in two different experimental paradigms: the wing imaginal discs and the abdominal histoblast

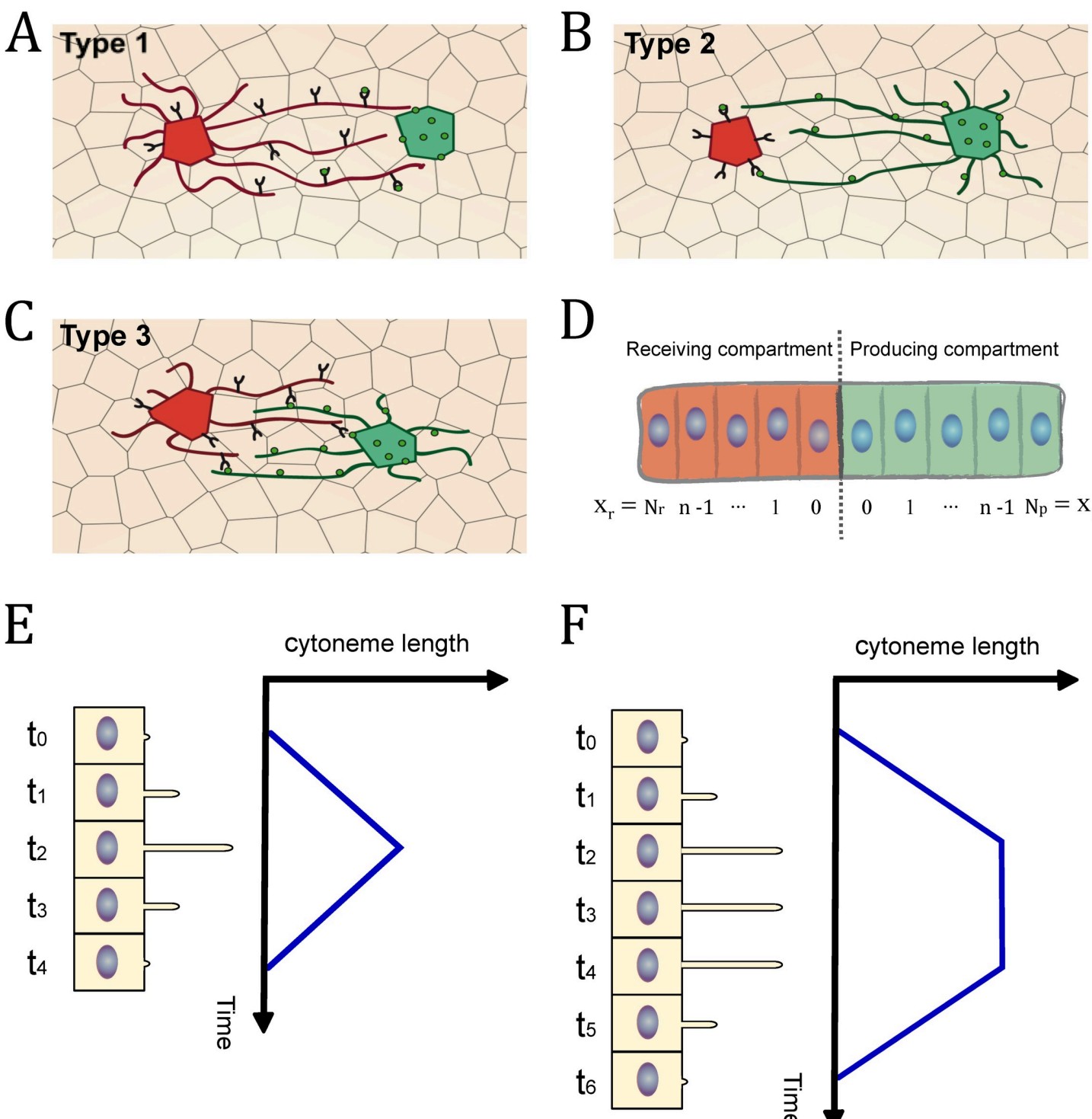

**Fig 1. Schemes of cytoneme-mediated cell signaling based on experimental evidence**: (A) Type 1: Receiving cells emit cytonemes to collect the morphogen from producing cells. (B) Type 2: Producing cells emit cytonemes to deliver the morphogen to receiving cells. (C) Type 3: Both producing and receiving cells emit cytonemes to deliver and collet the morphogen respectively. (D) Frame of reference used to develop the mathematical equations. (E) Schematic representation of the cytoneme triangular dynamics. (F) Schematic representation of the cytoneme trapezoidal dynamics.

nests of *Drosophila*. As for the adaptability of Cytomorph, we also made a preliminary analysis

and *in silico* predictions of Dpp and Wg gradients in wing discs.

## Results

### Theoretical framework: Mathematical model

The morphogen gradient distribution is usually studied in biology as a spatial 1D function that can be generally determined as:

$$\frac{\partial u(x,t)}{\partial t} = P(u,x,t) + T(u,x,t) - D(u,x,t) \qquad (\text{Eq} - 1)$$

Where $u$ is the concentration of a specific morphogen, $P(u,x,t)$ is the production term, $T(u,x,t)$ is the transport term and $D(u,x,t)$ is the degradation term.

The effect of morphogen production and degradation on the gradient shape has been described in some modeling works [6–9]. Here, we have focused on the transport mechanism; other terms were included using direct experimental data. To model cytoneme-mediated morphogen transport, we focused in the main requirement of this type of signaling; the establishment of cell-to-cell membrane contacts for localized transmission. Thus, the core of our mathematical model is based on the determination of cytoneme contacts, specifically the contact distribution at receiving cells. Therefore, the transport can be determined as:

$$T(u,x,t) = \alpha \cdot N(x_r,t) = \alpha \cdot \sum_{x_p=0}^{N_p} C(x_{p,r},t) \qquad (\text{Eq} - 2)$$

Where $N(x_r,t)$ is the total number of contacts in receiving cells taking into consideration all the producing cells involved ($N_p$). The $C(x_{p,r},t)$ is the contact function that defines the contact mechanisms for cytoneme signaling according to the position of producing ($x_p$) and receiving ($x_r$) cells.

Experimentally, three types of cell-to-cell contacts have been reported for cytoneme intercellular communication (Reviewed in [40,41]) (Fig 1A–1C):

Type 1: Cytonemes from receiving cells that contact signal-producing cell bodies (Fig 1A).

Type 2: Cytonemes from signal-producing cells that contact receiving cell bodies (Fig 1B).

Type 3: Cytonemes from both signal-producing and receiving cells that establish contacts (Fig 1C).

Developing tissues are usually compartmentalized into two cell populations that divide the morphogenetic field in signal producing and receiving areas. Therefore, it is common to describe gradients using a spatial 1D frame of reference (Fig 1D) in terms of the discrete cell positions ($x_{p,r} \in \mathbb{N}_0$) in these areas.

To generate a model for the cytoneme mediated signaling observed experimentally, three different contact functions $C_i(x_{p,r},t)$ are defined in terms of spatial conditions as follows:

Types 1 and 2: In order to establish contacts, the distance between cells must be smaller than, or equal to, the length of the cytonemes.

Type 3: In order to establish contacts, the distance between a producing and its receiving cell must be smaller than, or equal to, the sum of the lengths of the cytonemes extending from these cells.

Which mathematically can be represented, in our frame of reference, as:

- Type 1 : $C_1(x_{p,r},t) = \begin{cases} 0 \ \text{if} \ x_p \geq \lambda_r(t) - x_r \\ \psi(\mu,x_r) \ \text{if} \ x_p < \lambda_r(t) - x_r \end{cases}$ $\qquad (\text{Eq} - 3.1)$

- Type 2 : $C_2(x_{p,r}, t) = \begin{cases} 0 \ if \ x_p \geq \lambda_p(t) - x_r \\ \psi(\mu, x_r) \ if \ x_p < \lambda_p(t) - x_r \end{cases}$ (Eq − 3.2)

- Type 3 :

$$C_3(x_{p,r}, t) = \begin{cases} 0 \ if \ x_p < \lambda_r(t) - x_r \\ \psi(\mu, x_r) \ if \ \lambda_p(t) - x_r \leq x_p < \lambda_r(t) + \lambda_p(t) - x_r \\ 0 \ if \ x_p \geq \lambda_r(t) + \lambda_p(t) - x_p \end{cases} \quad (Eq − 3.3)$$

These equations describe the contacts between a receiving cell at position $x_r$ and a producing cell at position $x_p$, depending on the temporal dynamics of cytoneme length $\lambda_r(t)$ [$\lambda_p(t)$] and the probability of contact $\psi(\mu,x)$.

$\lambda_r(t)$ and $\lambda_p(t)$ describe the dynamics of elongation and retraction of cytonemes emanating from either receiving cell position $x_r$ or producing cell position $x_p$. Cytoneme dynamics [30] have shown that once cytonemes are initiated they do not only elongate and retract (named as Triangular behavior by [30]) (Fig 1E), but they can also have intermediate stationary phases, during which the cytonemes maintain their maximum elongation (named as Trapezoidal behavior by [30]). (Fig 1F). Therefore, in our model we mathematically defined $\lambda_r(t)$ [$\lambda_p(t)$] considering experimental data of the more basic cytoneme behavior (Triangular dynamics) but including the possibility of a stationary phase (Trapezoidal dynamics) (see S1 Text).

The function $\psi(\mu,x)$ determines if there are contacts between cytonemes satisfying the minimum distance condition in a specific place $x$ with a probability of $\mu$. This function takes the value of 1 in the case of establishing contact (see S2 Text). Since the cellular mechanisms to create a contact are beginning to be elucidated [42], but not fully understood, here we used a probabilistic approximation to the real mechanism of signal transfer.

## Computational framework: Model implementation

The general design of our model allows its application to different cytoneme-mediated morphogen gradients. Thus, to take full advantage of this approach, we created a Matlab-language-based software called Cytomorph, in which it is possible to simulate different experimental data as well as test various hypotheses *in silico*.

Cytomorph was designed to introduce different inputs (experimental data and variables under study), compute in different modules the dynamics of cytonemes and their contacts to finally plot morphogen gradient features (Fig 2 shows a general workflow of the software). For practical purposes, here we will focus on the inputs and outputs of the model. However, a detailed flow chart showing the different stages of the Cytomorph machinery can be found in Fig A in S3 Text.

**Cytomorph inputs.** Cytomorph inputs were divided into two sets (Fig 2A). The first set, which is loaded to Cytomorph via a spreadsheet (Fig 2A–1) encompasses the experimental distributions of cytoneme lengths as well as timing of elongation, retraction and stationary phases during cytoneme dynamics (See Table A in S3 Text). The second set, loaded via a graphical user interface (GUI) (Fig 2A–2, see also Fig B in S3 Text), comprises: 1) average experimental values (e.g. cell size within a tissue as well as elongation and retraction velocities), 2) parameters that are difficult to measure with current experimental techniques (e.g. the contact probability and the temporal contact dynamics) and 3) features or parameters without experimental data but which are required for morphogen simulations (e.g. the number of cells needed for gradient formation

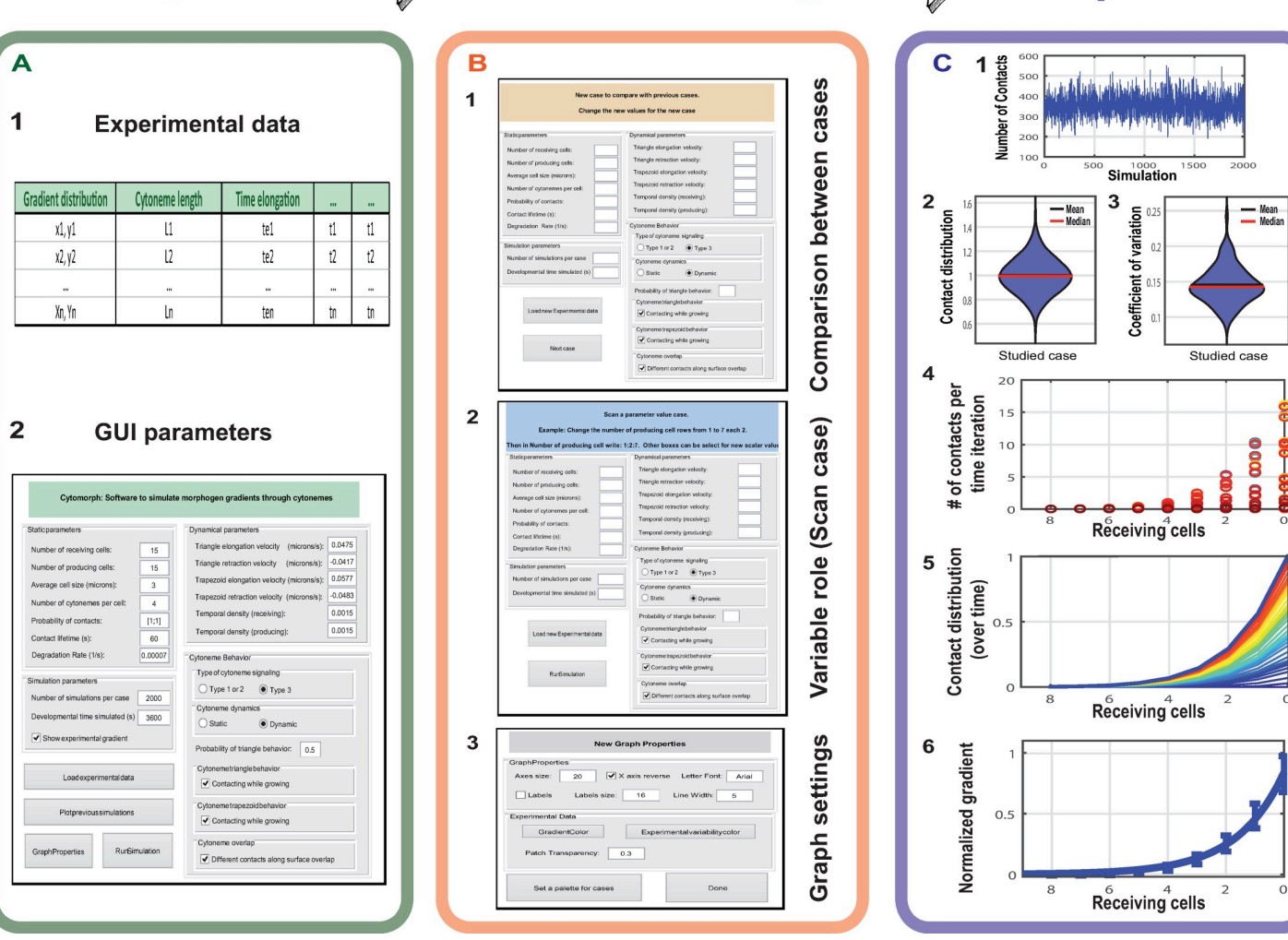

**Fig 2. General outline of the Cytomorph workflow:** (A) Inputs of the Cytomorph, divided into two groups: A.1) Loading the data through an Excel table. A.2) Loading it through the main GUI. (B) Cytomorph secondary GUI windows: B.1) Window in which different parameter combinations (cases) can be loaded to compare with the reference case. B.2) Window in which a scan of variable values can be selected to study their effect. B.3) Window in which graphical properties can be selected. (C) Graphic outputs of Cytomorph simulations: C.1 and C.2) Contacts per cell along simulations. C.3) Signal variability measured by coefficient of variation. C.4) Contacts per cell and iteration. C.5) Temporal evolution of the contact distribution. C.6) Final gradient and expected variability (error bars).

and the role of cytoneme dynamics or the frequency of cytoneme creation). A detailed illustration of features and parameters of the second group are described in Fig B in S3 Text.

**Cytomorph modules.** For an intuitive use of Cytomorph we designed a GUI to run simulations (Fig 2A and 2B) and Cytomorph was subdivided in different scripts and modules of the next three types: 1) A group of scripts for the GUI. 2) Modules to numerically simulate cytoneme dynamics and to compute contacts with their spatial distribution over time. 3) A module to plot the simulated contacts along different gradient properties (see S3 Text).

**Cytomorph outputs.** To study cytoneme features and assess their role in the Hh gradient formation, Cytomorph was implemented to analyze different characteristics:

- *Contact distribution*: The contacts per cell along simulations (Fig 2C-1); violin plots are shown (Fig 2C-2) to visualize the contact distribution along simulations.

- *Signal variability*: To study the predicted *in silico* variability we computed the distribution of coefficients of variation per case (Fig 2C-3).

- *Temporal evolution*: To observe the number of contacts in each receiving cell per time lapse (Fig 2C-4) and the total evolution of contact distribution and gradient shape over the simulated signaling time (Fig 2C-5).

- *Gradient distribution*: Assuming that each contact transmits the same amount of morphogen ($\alpha = const$ in Eq-2), the distribution of the morphogen $u(x,t)$ can be estimated through the $N(x,t)$ calculated in the model (Fig 2C-6).

A detailed description of how the outputs were computed and calculated can be found in Material and Methods and S3 Text.

## Experimental framework: Model validation

To validate Cytomorph we used the formation of the Hedgehog (Hh) gradient in two different *Drosophila* tissues: the imaginal wing disc and the abdominal histoblast nest. The latter has been used to study cytoneme dynamics as it easily allows *in vivo* imaging. Both tissues have the same cell distribution, in which the Hh producing region (Posterior (P) compartment) signals over the receiving region (Anterior (A) compartment).

We first characterized and quantified the biological magnitudes needed as inputs for Cytomorph simulations. For length characterization, we overexpressed Ihog, a trans-membrane protein and co-receptor of the Hh pathway present in all epithelial cells, since its overexpression stabilizes cytonemes without affecting their length [30]. This effect on cytoneme dynamics makes Ihog overexpression a good tool for cytoneme visualization at the basal side of fixed tissues, such as the wing imaginal disc (Fig 3A). Abdominal histoblast nests keep the same cell and cytoneme distributions as wing disc epithelia, with A and P compartments (Fig 3B) and apico/basal polarity (Fig 3C and S1 Movie). For wild type dynamics of cytonemes, we used markers that do not affect cytoneme dynamics (Life-actin-RFP and mCD8-GFP) that we simultaneously overexpressed using the binary systems UAS-Gal4 and QUAS-QF, which allows *in vivo* visualization of both, receiving (from A cells) and producing (from P cells) cytonemes (Fig 3C and S2 Movie).

Looking at the quantified length of cytonemes in the wing disc, we observed that receiving A cytonemes are significantly shorter than P cytonemes (Fig 3D). In addition, comparing both receiving and producing wing disc cytonemes with those of the abdominal histoblast nests [30], we observed that the former are significantly longer than the latter (Fig 3D). We also quantified the cell size ($\phi$) in both tissues and found a difference in cell size, with: $\phi = 3.05 \pm 0.65$ μm in wing discs and $\phi = 4.37\pm0.89$ μm in abdominal histoblast nests (see Materials and Methods for the measurement protocol and the statistical study of this average values).

To quantify the Hh experimental gradient (including its experimental signal variability) as a validation for the model-simulated gradient profile, we analyzed the signal intensity of endogenous Hh using a GFP fluorescent reporter (*Hh:GFP BAC*) in both *Drosophila* tissues. In addition, we also compared the Hh gradient responses between different samples (see Materials and Methods and S1 Fig for details) using a genetic tool (*EnhancerPtcRed*) that allows simultaneous visualization of Hh and the transcriptional response of its receptor Patched (Ptc) (Fig 4A). Following statistical analysis showed that, despite similar features between the tissues, the Hh gradients are not identical (Fig 4B), with the gradient decaying faster in abdominal histoblast nests than in the wing imaginal discs. It is important to mention that in our cytoneme-based model, the gradient shape is a consequence of the contact distribution (S2 Fig), which in turn, outcomes from the cytoneme dynamics and cytoneme distribution along the tissue.

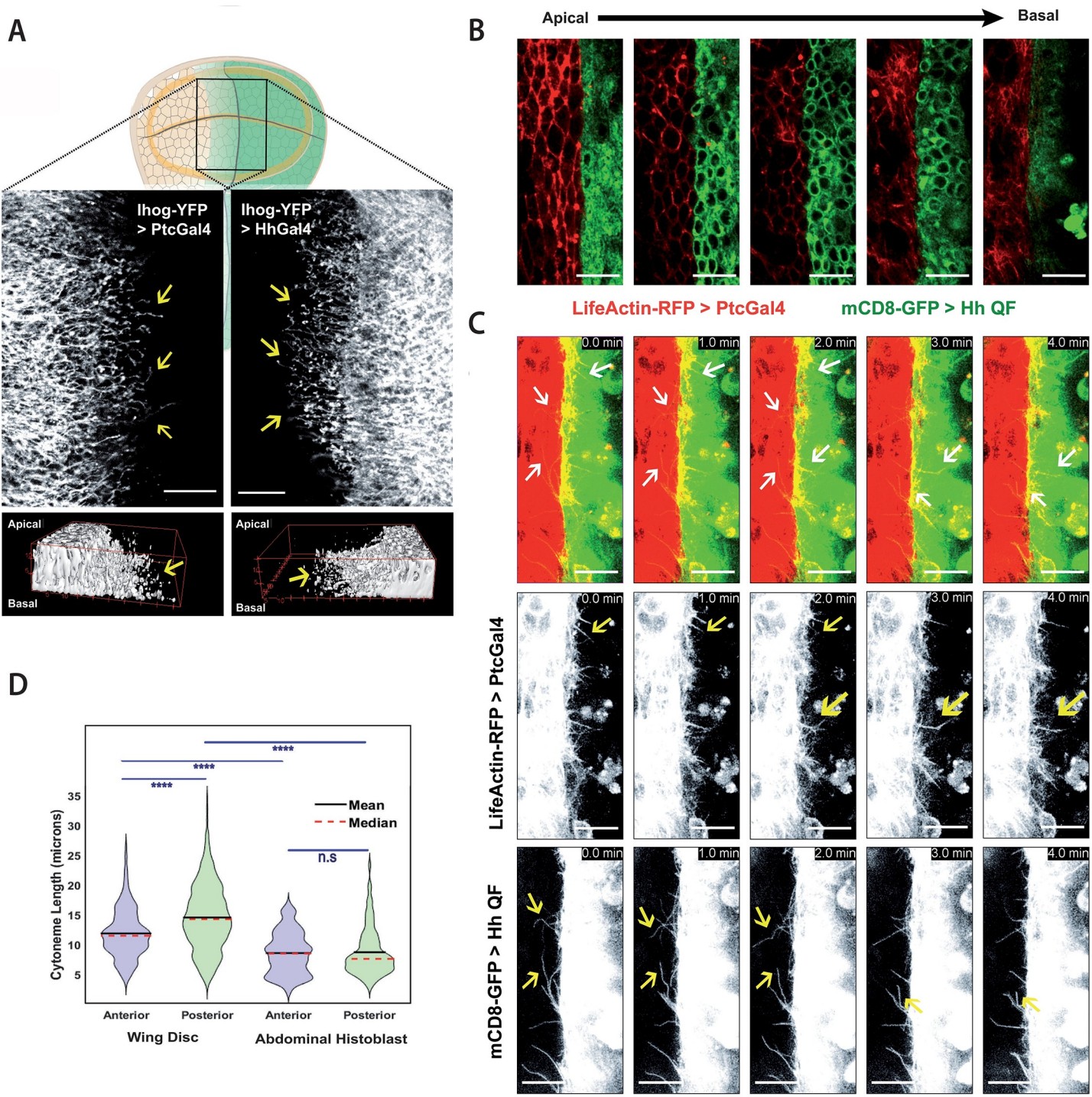

**Fig 3. Experimental cytoneme characterization in *Drosophila* tissues.** (A) Wing imaginal disc cytonemes protruding from A (top left) and from P (top right) compartment cells marked with Ihog-RFP. Bottom panels show 3D reconstructions of a confocal Z-stack taken at the basal side of the tissue showing cytonemes protruding and from A (bottom left) and P (bottom right) compartment cells. (B) A confocal Z-stack taken from the apical to basal side of the abdominal histoblast epithelium with the A compartment marked with life-actin-RFP (red) and the P compartment marked with CD8GFP (green). (C) *In vivo* temporal sequence of abdominal histoblast cytonemes taken at one-minute intervals. Top image sequences show both A and P compartment labelled cytonemes (A in red, P in green), middle image sequences show a single channel of A compartment cytonemes, and bottom image sequences show the single channel of P compartment cytonemes. (D) Statistical violin plots of cytoneme length distribution in the A (blue) and the P (green) compartments in wing disc (left) and abdominal histoblast nest (right). Scale bars: 15μm.

Furthermore, we observed that the Hh gradient range in both tissues can be determined by the sum of the maximum cytoneme lengths, emphasizing again the importance of cytonemes as mechanism for gradient formation.

This scaling of Hh gradients between the two tissues indeed provides an opportunity to study the adaptability of our cytoneme model. Analysis of the parameter space following loading of the experimental data (length, temporal dynamics of cytonemes and cell size $\phi$) within *in silico* simulations showed that our model is truly able to predict the shape of Hh gradients in the two tissues (blue fitted curve in Fig 4C and 4D). Thus, we demonstrate the power of our model to adapt to different biological conditions and correctly forecast the signaling gradient. The parameter space has been selected as a reference case for further simulations (S1 Table). Furthermore, when analyzing the parameter space used to fit the gradient, additional information can be deduced: for example, model simulation for the Hh gradient in the wing disc agrees with the cytoneme contact type 3, which fits with the experimental data. Modeling then, emphasizes the importance of this direct cytoneme-cytoneme (cyt-cyt) interaction towards the wing disc gradient correct development, so far assumed but not demonstrated. As for the gradient in the abdominal histoblast nests, a lower probability for cyt-cyt interaction fits better with the experimental gradient, indicating that, in contrast with the wing disc tissue, in abdominal histoblasts, cyt-cyt interaction is not as critical as cytoneme to cell body contact.

Since many theoretical models still assume free diffusion as the mechanism for morphogen transport [10], we next compared our cytoneme model predictions with those of the classical diffusion-degradation model (see Materials and Methods), and then both with the experimental gradient measurements. Interestingly, in the case of the wing disc, both model predictions fall statistically within the experimental variability (Fig 4E), with the cytoneme model slightly closer to the experimental mean. However, this is not the case for the abdominal histoblast nests where the diffusion model does not adapt and is not able to predict the gradient as accurately as the cytoneme model does (Fig 4F, black). Indeed, the diffusion model requires the assumption of a three times smaller diffusion constant to fit simulations with experimental data in abdominal histoblast nests (Fig 4F, red).

### *In silico* framework: Model predictions

Hence, the model can be used to study the effect of different parameters on gradient formation and to predict their biological implications. As examples, we selected two parameters from which we could acquire experimental data (cytoneme length/cell size ratio and number of producing cells) as well as a parameter lacking quantified experimental data (number of cytonemes per cell). We then used Cytomorph simulations to predict their effects and values (Fig 5).

- *The ratio between cytoneme length and cell size* is a default unit used in the software to intuitively visualize the extent of cytonemes. *In silico* simulations showed that this ratio seems to be responsible of controlling the shape and length of the morphogen gradient (Fig 5A). Although this ratio also affects signal variability, this is not statistically significant in most cases (Fig 5A'). Therefore, after simulation we can conclude that both, length of cytonemes and cell size, are key to understand how cytoneme signaling defines the shape and extension of the gradient (Fig 5A").

- *The number of signaling source cells* has been previously suggested to be crucial for plausible gradient formation modeling [39], while experimental data also hints towards the importance of this trait for Hh signaling [43]. Thus, we decided to analyze the effect of this parameter in our model. We have observed that, starting count from the A/P compartment border,

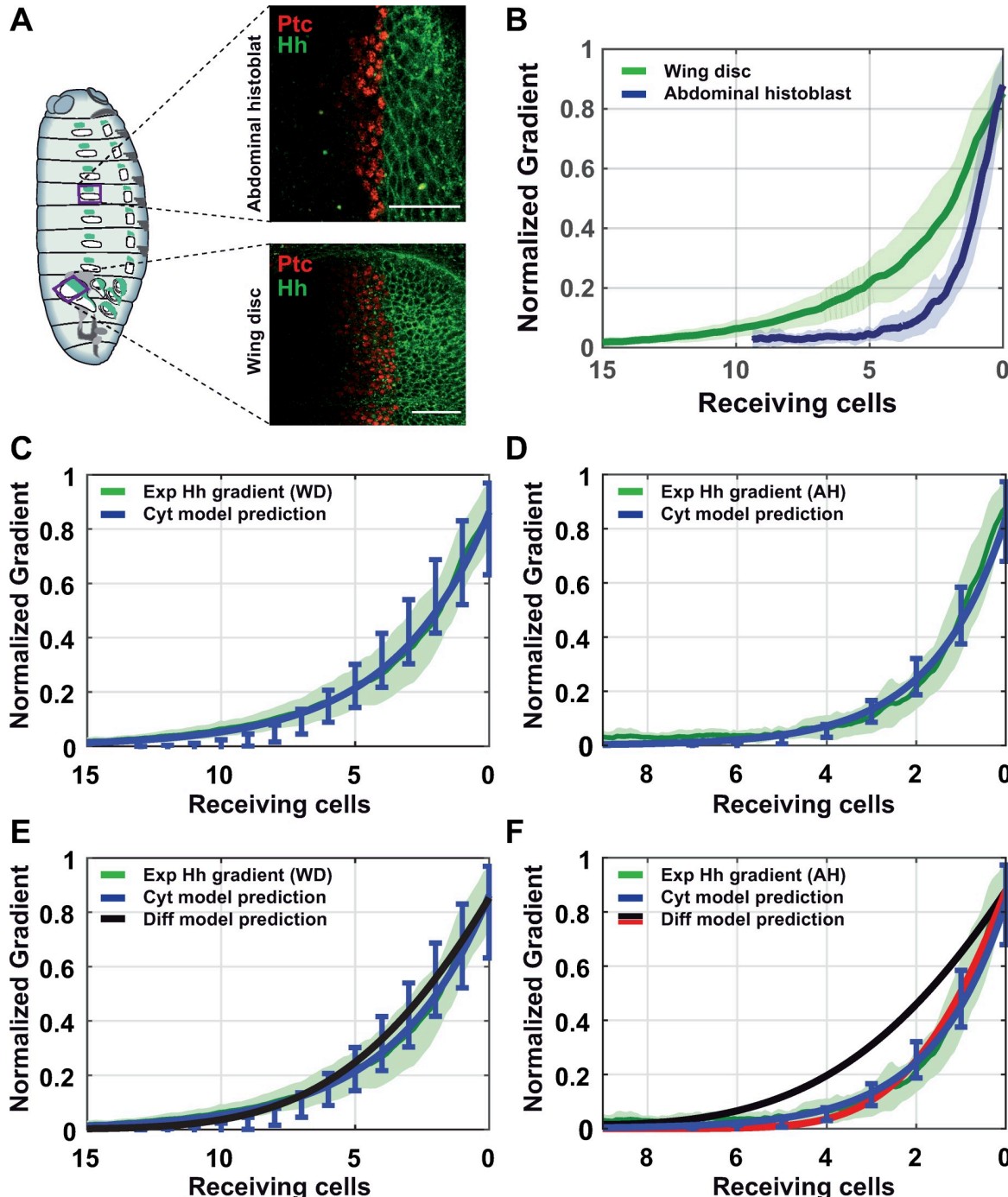

**Fig 4. Experimental and simulated Hh gradients in *Drosophila* tissues.** (A) Confocal sections of *Drosophila* epithelia labeled with *Hh: GFP BAC* and *EnhancerPtcRed*. Top: abdominal histoblast nest. Bottom: imaginal wing disc. (B) Quantified data of the Hh gradient in both epithelia: wing disc (green) and abdominal histoblast nest (blue). (C) Comparison between the wing disc experimental gradient (green) and the predicted gradient estimated by our cytoneme model (blue). (D) Comparison between the abdominal histoblast nest experimental gradient (green) and the predicted gradient estimated by cytoneme model (blue). (E) Comparison between the wing disc experimental gradient (green) and the predicted gradients applying different models: cytoneme model (blue) and diffusion-degradation model (black). (F) Comparison between the abdominal histoblast experimental gradient (green) and the predicted gradients applying different models: cytoneme model (blue) and diffusion-degradation model with different diffusion coefficient (red 3 times smaller than black). Scale bars: 30μm.

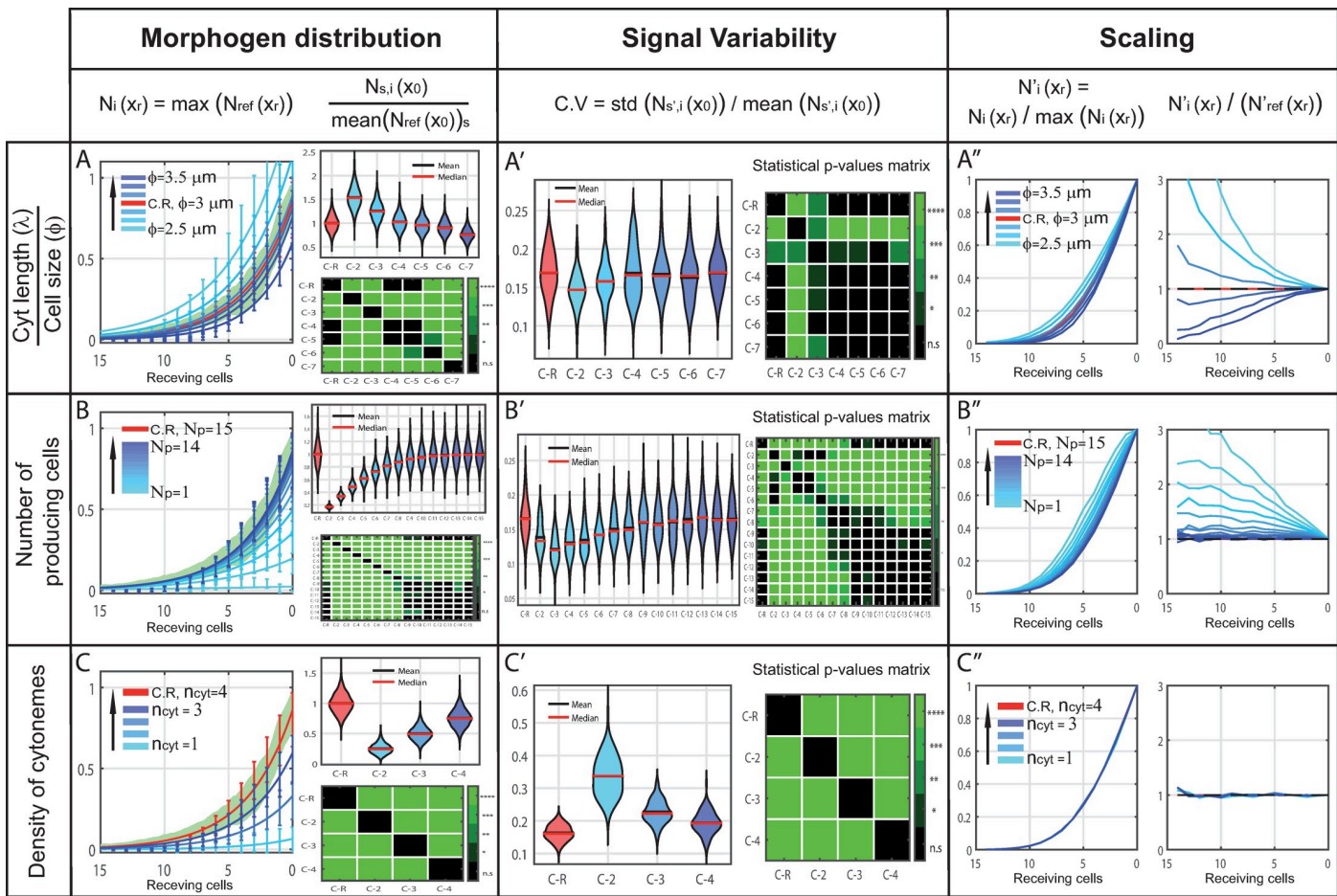

**Fig 5. *In silico* study of different cytoneme variables and their predicted impact on gradient features.** Reference simulation in red, simulations after modifying a specific parameter in blue (graded light to dark depends on the value) and experimental data in green. X) Left. Morphogen distribution for different cases, normalized to the maximum value of the reference case, along receiving cells including the expected variability per cell row (error bars). Right. Study of the number of contacts in the first row of receiving cells $x_0$, normalized to the average value of the reference case: top, violin plots of 2000 simulations per case; bottom, green-color-coded matrix of p-values for the violin distributions. X') Coefficient of variation per case in the first row of receiving cells $x_0$ (left). Green-color-coded matrix of p-values for violin distributions (right). X") Distribution of contacts normalized to their maximum value to compare changes in gradient shape along receiving cells (left). Coefficient of the normalized distributions to study the scaling along receiving cells (right). (A) Simulations for different cell size/cytoneme length ratios ($\phi$ = 2.5 to 3.5 each 0.2 μm (blue), $\phi$ = 3 μm (red)). (B) Simulations for different number of producing cells rows involved in the signaling (Np = 1 to 14 (blue), Np = 15 (red)). (C) Simulations for different number of cytonemes per cell (ncyt = 1 to 3 (blue), ncyt = 4 (red)).

the first 5–7 rows of producing cells are key in shaping the Hh gradient (Fig 5B), while the next cell rows (rows 8–10) refine the gradient shape lowering the variability, and further cell rows (rows > 10) does not affect the Hh gradient. This can be observed in both the amount of morphogen and the signal variability (Fig 5B and 5B'). Therefore, there is a key number of producing cell rows indeed important in the scaling and determination of the gradient shape in the wing imaginal disc (Fig 5B"). This dependence on producing region size can be extrapolated to other tissues, as suggested by our simulations in the abdominal histoblast nests (S3 Fig). In contrast, analysis of the number of receiving cell rows did not show any effect on Hh distribution (see S4 Fig).

- *The density of cytonemes* (the number of cytonemes per cell) is key for the amount of morphogen distributed (Fig 5C) and significantly affects signal variability (Fig 5C'), but it is not a determinant factor for the gradient shape (Fig 5C"). Counter-intuitively, experimental

variability can be estimated *in silico* (error bars versus green shaded area in Fig 5A) with a low number of cytonemes per cell, and this can also be inferred from the experimental wild type (Fig 3C). These results are biologically significant as they strongly suggest that the gradient shape is mainly determined by cytoneme behavior and not by their number.

## Hypotheses based on cytoneme behaviors

Cytomorph is an adaptable tool devised to answer different questions and test hypotheses on cytoneme mediated signaling. Since most of our working hypotheses are related to contact dynamics, we will next focus on how the three types of contacts (Fig 1A–1C) might affect gradient features. Types 1 and 2 can be considered mathematically the same, while type 3 should have an additional contribution, as both presenting and receiving cells emit contacting cytonemes. Our model predicts a significant effect in the gradient when considering the type 3 cytoneme contacts compared with those of types 1 or 2, as we found that the amount of morphogen (Fig 6A) and the length of the signal (Fig 6A") were doubled. Type 3 seems then to be the most probable situation for Hh gradient formation (Fig 6A), although types 1 and 2 can still be functional forms for other signaling situations.

Further analysis of contact dynamics properties using Cytomorph has also allowed the study of the contact effect for reception, which is currently not well understood due to the difficulty of approaching it experimentally; it was defined in the model by the contact probabilistic function $\psi(\mu,x)$. In particular, we implemented in different Cytomorph modules three working hypothesis of the contact function $\psi(\mu,x)$:

1. A contact dynamic in which the probability of contact only depends on the condition that cytonemes are close enough; then the probability to contact is $\psi = \psi(\mu)$.

2. A contact dynamic in which different contacts can be established along the overlapping cytoneme membranes. This multiple-contacts approach was designed by the special contact function $\psi = \Psi(\mu)$.

3. A contact dynamic in which, in addition to the previous distance condition, the cell position is important and can be treated as a variable $= \psi(\mu,x)$.

Comparing the first two hypothesized contacts, the *in silico* simulations showed that the overlapping multiple contacts function can significantly change the number of contacts and subsequently the amount of morphogen transferred (Fig 6B), also resulting in significant changes over signal variability (Fig 6B') and gradient shape (Fig 6B"). Similarly, comparison between cases 1 and 3 showed statistically significant changes in the number of contacts (amount of transmitted morphogen) and signal variability when including cell position as a variable (Fig 6C and 6C'). Furthermore, after the analysis of different scaling across receiving cells we could also infer that gradient distribution was affected (Fig 6C"), with case 3 showing a faster and more linear decay as a consequence of its dependence on cell position (S2 Text).

Since our results suggest that the contact probability function only depends on the variable $\mu$, we carried on an *in silico* study to test the impact of this variable over gradient features. Interestingly, model simulations for different values of $\mu$ only significantly contribute to the amount of morphogen transferred (Fig 6D), but they do not disturb neither the variability nor the shape of the morphogen gradient (Fig 6D' and 6D").

## Cytoneme dynamics in Hh gradient evolution

To this point we have validated Cytomorph in steady state conditions (Fig 5), and from now on we will test its capability to study temporal aspects during gradient formation. For this

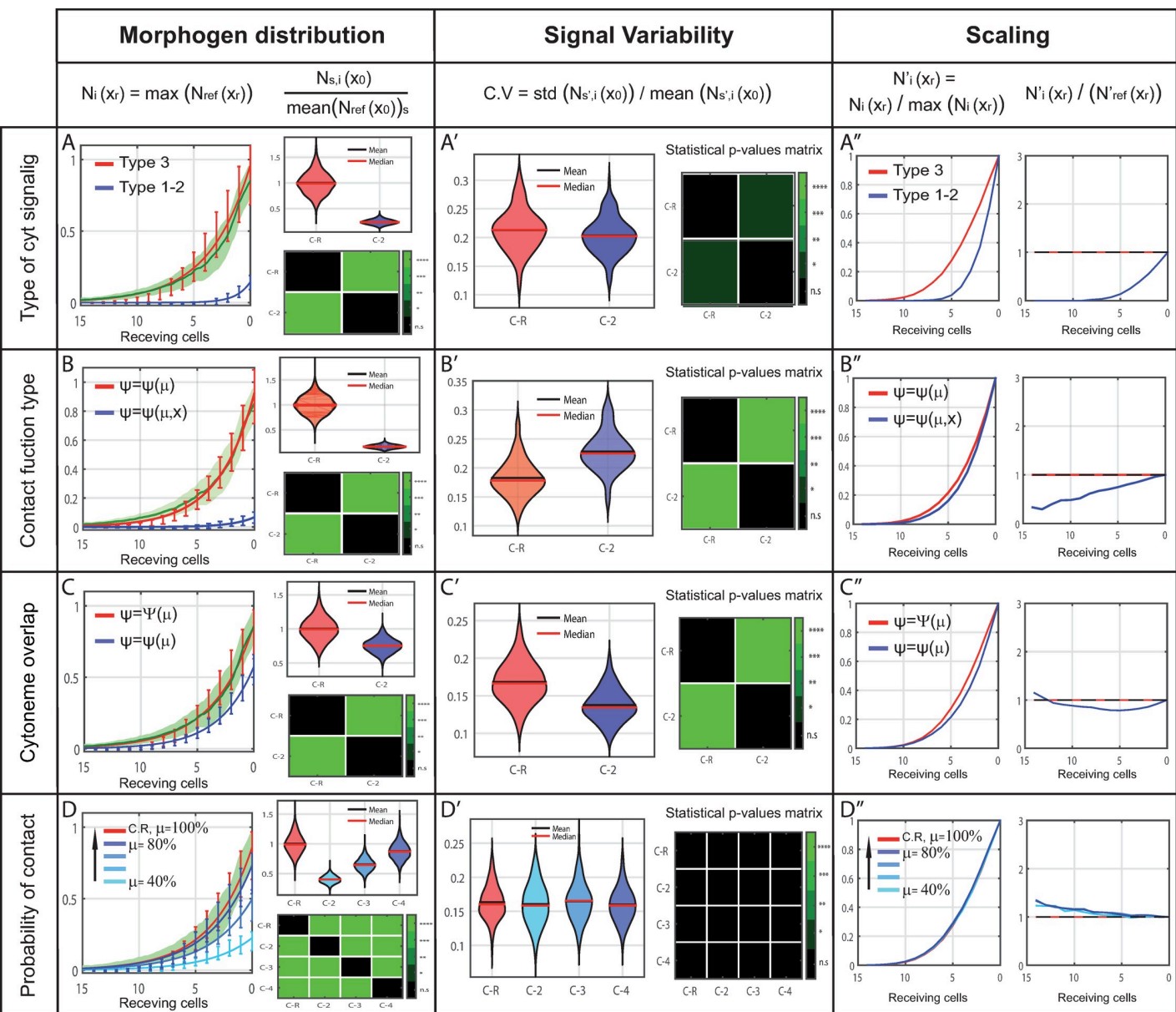

**Fig 6. *In silico* study of different cytoneme presumptions and their predicted impact on gradient features.** Reference case in red, simulations after modifying a feature in blue and experimental data in green. X) Left. Morphogen distribution along receiving cells for different cases, normalized to the maximum value of the reference case, showing the expected variability per cell row (error bars). Right. Study of the number of contacts in the first row of receiving cells $x_0$, normalized to the average value of the reference case. Top, violin plots of 2000 simulations per case. Bottom, green-color-coded matrix of p-values for the violin distributions. X') Coefficient of variation per case in the first row of receiving cells $x_0$ (left). Green-color-coded matrix of p-values for violin distributions (right). X´´) Distribution of contacts normalized to their maximum value to compare changes in gradient shape along receiving cells (left). Coefficient of the normalized distributions to study the scaling along receiving cells (right). (A) Simulations for different cytoneme signaling type (type 3 in red and type1-2 in blue). (B) Simulations for different contact functions (type $\psi(\mu)$ in red and type $\psi(\mu,x)$ in blue). (C) Simulations of the hypothetical case of multiple contacts between cytonemes along the overlapping surface (single contact in blue, multiple contacts in red). (D) Simulations for different probability of contact ($\mu = 40\%$ to $80\%$ each $20\%$ in blue, $\mu = 100\%$ in red).

purpose, we performed fluorescence recovery after photobleaching (FRAP) experiments in the abdominal histoblast nests. In this tissue, the gradient is established previous to histoblast migration and it allows the dynamic characterization of the Hh signaling gradient by *in vivo* recording (S3 Movie).

Previous to photobleaching a reference Z-stack was taken and the signal was then bleached to 80–90% of the initial maximum value (Fig 7A); recovery was then recorded in a Z-stack every 45 seconds. To automatize the acquisition of the gradient profile, a FIJI macro was

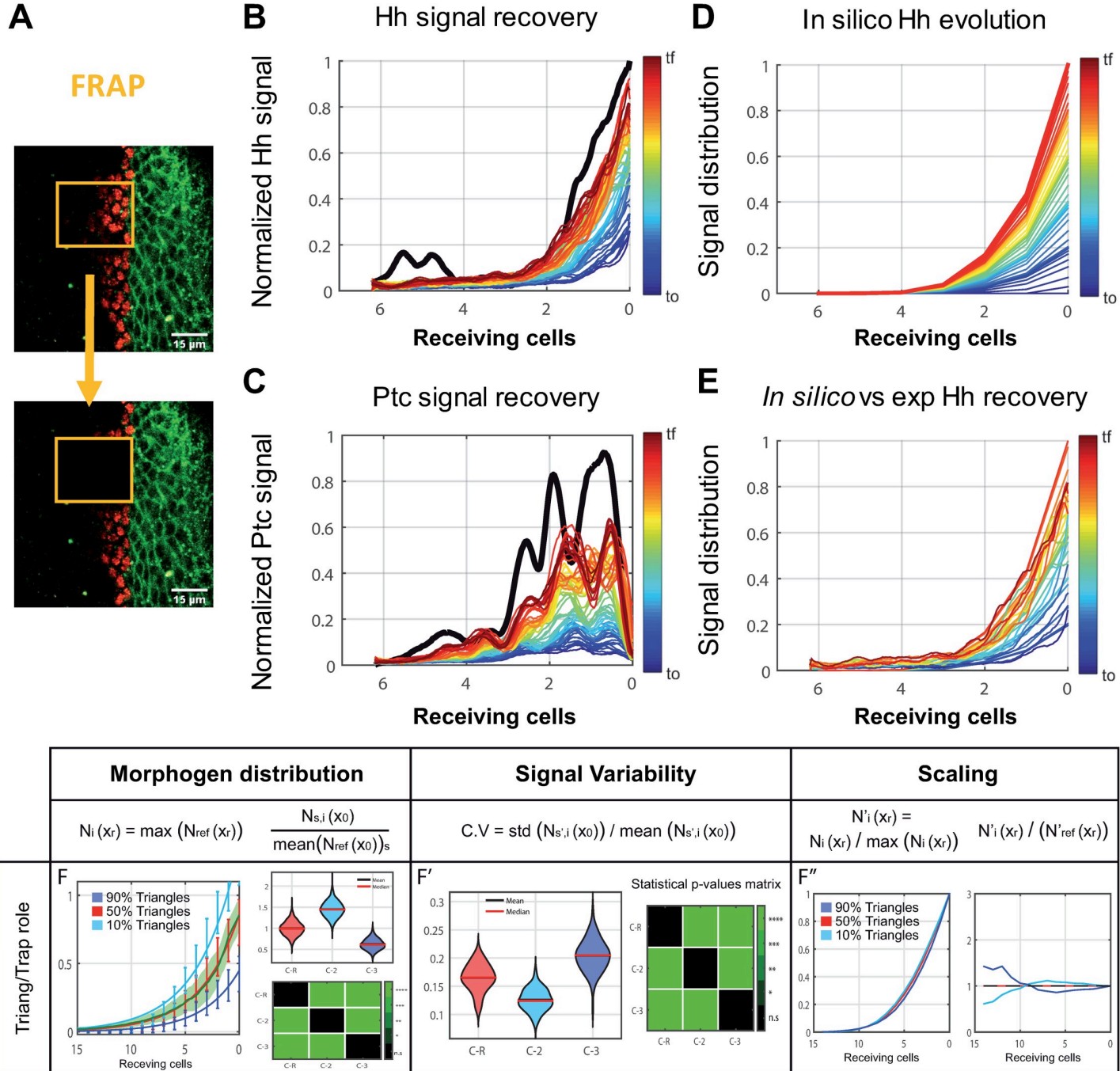

**Fig 7. FRAP experiments to study the temporal gradient formation.** (A) Representative image of FRAP experiments in abdominal histoblast nests in which the signal is eliminated after photobleaching over a specific ROI. (B) Hh (*Hh:GFP BAC*) gradient profile shortly before bleaching in black and Hh signal recovery over time coded in a hot colormap; each step corresponds to 45 seconds. (C) *Ptc* (*EnhancerPtcRed*) expression profile shortly before bleaching in black and *ptc* signal recovery over time coded in a hot colormap, each step is 45 seconds. D) *In silico* signal evolution predicted for abdominal histoblast nests. E) A graphical comparison every 3 minutes between *in silico* simulations and experimental data. F) Simulations for cytonemes contacting while growing with a different proportion of triangular/trapezoidal cytoneme dynamics (10% triangles in light blue, 50% triangles in red, 90% triangles in dark blue). Scale bars: 15µm.

written. The results showed that the Hh gradient recovered up to 92% of the initial value in less than 50 minutes (Fig 7B), while the receptor graded response (*EnhancerPtcRed*) presented a 64% recovery in the same time. (Fig 7C). This difference in the percentage of recovery was expected, as there is a delay in the reporter response, which requires both transcription and translation. To validate the temporal evolution of the model, we then simulated the Hh gradient in histoblast nests using Cytomorph (Fig 7D) and compared the predicted with the experimental curves. As we can observe in Fig 7E, the temporal prediction and the experimental signal recovery are the same, corroborating the capacity of Cytomorph to predict the temporal evolution of the gradient.

Cytomorph could also be used to solve pending questions regarding cytoneme dynamics. We were particularly intrigued by how each type of the observed dynamics [30] might affect the shape of the gradient. To analyze whether each cytoneme behavior represented or not an advantage for the distribution of the gradient and its variability, we modified the fraction of the cytoneme population having each behavior and simulated how these changes could affect the gradient (Fig 7F and 7F'). It was interesting to find out that the impact was stronger on the tail of the gradient when considering that cytonemes simply elongate and retract (triangular behavior), while the impact was greater near the source of morphogen (Fig 7F") when considering that all cytonemes have a stationary phase between elongation and retraction (trapezoidal behavior).

Although the experimental data showed the importance of the dynamics of cytonemes for the formation of gradients [22,23,28–30], we also included in the model the case in which all cytonemes were static. In fact, our simulations showed that dynamic and static cytonemes generate different gradient shapes (S5 Fig). This is noteworthy since many theoretical models do not consider this temporal aspect and our results indicate that using dynamic cytonemes the predicted gradient better fits the experimental data (S5 Fig). It should be remembered that the temporal dynamics of the contacts is not yet well defined and, furthermore, it is experimentally difficult to unravel how the dynamic contacts coordinate with growth during gradient formation. Cytomorph can be used to study plausible biological scenarios for Hh signaling, even in the absence of experimental data. In fact, simulations of contacts and growth (S5 Fig) allow to study the dynamics of signaling and reflect the complexity in the coordination of different cytoneme features in the formation of a gradient.

### Robustness of cytoneme signaling

Although failures can occur during the development of organisms, cell signaling has been shown to have a robust control mechanism [44–46] in which a combination of parameters can compensate for a possible developmental failure. In this context, Cytomorph can also serve to identify compensation mechanisms; as an example, we have found an interesting interaction between the number of producing cells and the density of cytonemes per cell: a reduction in the number of producing cells can be compensated by an increase in the number of cytonemes per cell (S6 Fig). Both predicted gradients fall within the experimental variability, creating a "functional" gradient that should be able to activate the same target genes.

### Cytomorph as a predictive tool for other morphogen gradients

We propose Cytomorph as a computational tool to simulate the formation of cytoneme-mediated gradients for different morphogens. Thus, to test this hypothesis we made a simple approximation using Cytomorph to simulate Dpp and Wg signaling pathways in the wing imaginal disc.

The graded distribution of Dpp signal in *Drosophila* wing disc has been extensively studied [47–52] and described to be mediated by cytonemes [16,53]. In this system, Dpp is produced by the initial rows of A compartment cells (~ 7 cell rows) and spreads across A and P compartments of the wing pouch, forming two morphogenetic gradients.

Using known characteristics of cytonemes in the imaginal wing disc, we simulated the graded distribution of Dpp within P compartment cells and compared it with the experimental published data [50]. The resulting simulation correctly predicts the real Dpp gradient shape (S7 Fig), besides enabling the extrapolation of some predictions for Dpp cytoneme behavior. Specifically, Cytomorph simulations suggest that for Dpp signaling as in the case of Hh, cytonemes elongate from both, producing and receiving cells, since this setup reproduces the described Dpp gradient extension and shape [50]. However and differing from the Hh case, our simulations also predict that Dpp cytonemes are not likely to establish contacts while growing, establishing contacts after elongation instead (S7 Fig).

Although the Wg gradient has not yet been shown to be mediated by cytonemes in the wing disc, there are strong evidences of for cytonemes-mediated Wg signaling in other *Drosophila* tissues [18] and in a vertebrate system [19]. In the wing disc, Wg signaling fulfills the cell distribution assumed by Cytomorph: cells at dorso/ventral (D/V) compartment border of the wing pouch are the source of Wg, which spreads and signals to both D and V compartments. Thus, and using cytoneme length data for the imaginal wing disc, we were able to estimate the length for the Wg gradient in 42 μm (~ 14 cells) according to eq-3. This prediction is in agreement with experimental data for the extension of the Wg signaling gradient, described by the extracellular detection of the Wg protein in imaginal wing discs up to 40 μm [54].

## Discussion

In this work, we present a general *in silico* model for morphogen gradient formation which considers the cytoneme-mediated dispersion of morphogens. In particular, we demonstrate that this *in silico* model validates the cytoneme-mediated Hh gradient formation, and that this can be extrapolated to other morphogens such as Wg and Dpp. We have implemented our model into an open computational software (Cytomorph), which allows the introduction of experimental data to study the role of different biological parameters. With this approach we overcome nonexistent link between theoretical models and experimental data in cytoneme mediated cell signaling. To improve our understanding of how specific cytoneme features can impact the gradient properties, Cytomorph is capable of plotting results in graphs that show the final shape of morphogen distribution, as well as the number of contacts, the signal variability, the time course and the gradient scaling. To facilitate the use of this tool, we also designed a GUI that allows straightforward control of software commands.

### Cytomorph validation

To experimentally validate Cytomorph and its adaptability to predict real gradients, we studied the Hh gradient formation in two different *Drosophila* tissues: wing imaginal discs and abdominal histoblast nests. Using different genetic tools, we experimentally quantified several parameters in both tissues, such as the length of cytonemes, the cell size and the Hh distribution. Cytomorph was able to predict the Hh scaling and correctly simulate the signal gradients in both tissues, emphasizing the involvement of cytonemes for correct signaling. Although the quantified gradient scaling in these two tissues had not been previously characterized, we expected them to be different because, despite their similarities, both systems have different growth dynamics: while the wing disc is an expanding but static epithelium, the abdominal

histoblasts divide and migrate simultaneously reducing, for instance, the probability of cyt-cyt contacts, as our model suggests.

## Cytoneme model versus diffusion model

The diffusion model is still the mathematical model most commonly used in biophysics, although the biochemical properties of most morphogens argue against their transport via Brownian motion. Comparing our cytoneme model with the classic diffusion-degradation model, we found that our model thoroughly predicts the shape of the Hh gradient in two different tissues, the wing imaginal discs and the abdominal histoblast nests. The diffusion model, however, required a readjustment of the diffusion constant to predict the Hh gradient in the abdominal histoblast nests, despite being the same protein in similar epithelial tissues. Nevertheless, it is important to point out that measurement of the diffusion constant is an effective parameter that summarizes a collective behavior and does not give information regarding the transport mechanism involved. It has been stated indeed, that diffusion coefficients can significantly vary depending on the morphogen, the tissue and the experimental approach [55].

## *In silico* study of cytoneme features

After experimental validation of Cytomorph, we then studied different aspects of cytoneme-mediated signaling *in silico*, as a way to better understand the specific role of cytoneme features, and enable hypotheses generation regarding this signaling mechanism. We first tested the cytoneme length/cell size ratio, a parameter for which we already had experimental data. Our simulations suggested that this is a crucial parameter for Hh gradient scaling, but not for the signal variability. This model prediction was experimentally supported in abdominal histoblast nests and in imaginal wing discs.

Theoretical analysis [39] emphasizes the importance of considering an extended source to predict realistic gradients, however previous models do not take this element into account. Therefore, we used Cytomorph to clarify the effect of an extended source in shaping the gradient. The resulting simulations gave a detailed description of how the gradient is affected by changing the number of cell rows involved in the production of morphogen in a tissue.

In addition, Cytomorph also allows analysis for the potential effect over gradients of parameters for which no experimental data is available, such as the number of cytonemes per cell. Interestingly, our results suggest that this particular parameter is key for both the variability of the signal and the amount of transmitted morphogen but not for the distribution or scaling of the gradient. Moreover, simulations also estimated the likelihood of a low number of cytonemes per cell.

In parallel to cytonemes parameters, we have also studied other features of cytoneme-mediated signaling. Thus, by observing the effect of different types of signaling contacts, our model predicts that the type 3 is different from types 1 and 2, since the amount of transmitted morphogen and the length of the gradient increases due to cyt-cyt contacts. Furthermore, in agreement with experimental observations, our *in silico* results also showed that type 3 cytoneme interaction is the most likely situation for Hh gradient formation in the wing imaginal disc.

Previous approaches to cytoneme signaling used weight functions, with a dependence on cell position, to ponder the quantity of received morphogen. To ascertain if this dependence is required, we tested three different hypotheses for the contact function $\psi(\mu,x)$. In contrast with previous approaches, our model suggests that this contact probability does not require a pondered mechanism based on cell position, since the simplest case $\psi = \psi(\mu)$ fits the gradient distribution better than $\psi = \psi(\mu,x)$. Besides, the existence of multiple contacts between cytonemes ($\psi = \Psi(\mu)$) fits better the experimental data for Hh signaling. Since Cytomorph predicts a low

number of cytonemes per cell, then the key feature to determine the gradient shape is the distribution of contacts across receiving cells. This conclusion is as well in agreement with results for other cytoneme-mediated morphogen distributions [56].

Finally, we studied the effect of the probability of contact $\mu$ over gradient properties. Our results showed that this parameter can significantly impact the amount of morphogen transmitted but not the signal variability or the gradient shape.

### Cytoneme dynamics in Hh gradient formation

One of the main advantages of our model is the inclusion of the temporal dynamics in the equations, a feature that has been experimentally found to play a significant role for the correct activation of target genes during development. Thus, to validate the temporal dynamics of our model we performed experiments to study the timing of Hh signal recovery after photobleaching (FRAP technique), comparing these data with our model simulations. Comparison of the predicted and the experimental gradient curves proved that our dynamic model is indeed able to simulate physiological temporal features.

We then used Cytomorph to study the potential role of the observed two types of cytoneme behavior (Triangular and Trapezoidal). Our simulations showed that the two cytoneme dynamic might have a distinctive impact on specific regions of the Hh gradient, indicating that these cytoneme behaviors are important for the precise spatial control of the gradient shape. In addition, our simulations also revealed that a population half Triangular and half Trapezoidal for cytoneme dynamics fits better the experimental data, a proportion found too experimentally [30].

The simulations indeed point out that static and dynamic cytonemes would give rise to quite different Hh gradients. Consequently, this characteristic of cytoneme signaling should be included in theoretical models that study cytoneme signaling. Moreover, cytoneme temporal dynamics could provide robustness to the progressive establishment of signaling gradients, an advantage for both growing (wing imaginal discs) and migrating while growing systems (abdominal histoblast nests). Static cytonemes are less likely to adapt to tissue changes, increasing the probability of failure, while dynamic cytonemes can allow constant regulation of the gradient shape throughout development. Nevertheless, other static tubular structures, such as tubulin-based channels, could be significant for other morphogens or biological models [57].

Our *in silico* model emphasizes the role of different features for the gradient properties. Particularly, an important prediction is that the gradient shape is a consequence of the contact distribution; which in turn is due to cytoneme dynamics and cytoneme distribution along the tissue. In fact, the model suggests that for the correct establishment of graded distribution, cytoneme dynamics are more critical than the amount of morphogen available from producing cells. This hypothesis has been recently corroborated experimentally for several morphogens. The analysis of Hh, Wg, and Dpp dispersion in the Drosophila wing imaginal disc indicates that their delivery to target cells must be regulated, since an increment in their gene doses does not alter the extent or shape of their gradients [58]. For Hh signaling, the receiving cells take up less than the 5% of Hh produced; under conditions of Hh production up to 200% of the normal amount, neither the protein uptake nor the extent of the gradient changes. These findings agree with a regulated mechanism for morphogen delivery, such as the cytoneme-mediated model, and not with a diffusion model.

### Other uses of Cytomorph

Based on uncomplicated mathematical premises, our model improves the understanding of cytoneme signaling mechanisms. Although our interest along this work has been focused on

identifying individual roles for different cytoneme parameters and their effects over the morphogen gradient formation; Cytomorph was also useful to detect cytoneme features interactions able to counteract malfunctions, emphasizing the robustness of the cytoneme model as a signaling mechanism.

Finally, we propose that Cytomorph can adapt to different biological systems and morphogens. We used Cytomorph to simulate Dpp and Wg signaling, as an initial attempt to verify its use for other morphogens. We showed that the simulated Dpp gradient reproduces the experimental gradient and that the length of Wg gradient can accurately be estimated with our model equations. However, further in detail studies are required since each morphogen and tissue has its own characteristics. For instance, in both, Wg and Dpp gradients, some of the signal-producing cells function also as receiving cells, different from the case of Hh. This exemplifies a case where the software plasticity would allow the implementation of additional computational modules. The adaptability is achieved by the modular architecture used to design Cytomorph that can adjust to the particularities of each system and the testing of emerging biological hypotheses.

## Material and methods

### Experimental methods

***Drosophila* lines.**   *Drosophila melanogaster* stocks were maintained according to protocols described in Ashburner manual [59]. Crosses were maintained at 18˚C until the time of gene expression induction The description of mutations, insertions and transgenes is available at Fly Base (http://flybase.org).

The following drivers were used to induce ectopic expression using the Gal4/UAS [60] and QUAS-QF [61] systems: tubGal80[ts] (Bloomington *Drosophila* Stock Center, BDSC), *hh.Gal4* [62], *ptc.Gal4* [63] and *Hh-QF* (generated by Ernesto Sánchez-Herrero, CBMSO).

Overexpression stocks: The *pUAS*-transgene strains used were: *UAS.ihog-YFP* [64] *and UAS.LifeActinRFP (BDSC 58362)*. The QUAS-transgene strains used were: *QUAS.mCD8-GFP* (BDSC 30002).

Other stocks: *EnhancerPtcRed* (Kyoto stock center, DGRC 109138) *and Hh*:*GFP BAC* [65].

### Experimental data acquisition and quantification in wing imaginal discs

Laser scanning confocal microscopes (LSM700 and LSM800 Zeiss) were used for confocal fluorescence imaging of imaginal discs. Fluorescence signal of *Hh:GFP BAC* protein and *EnhancerPtcRed* reporter were obtained using 40 x magnification and taking Z-stacks with a step size of 0.7–1 μm. Fiji software (ImageJ software, National Institutes of Health) was used for image processing and analysis.

**Filopodia extension.**   Cytonemes were labeled overexpressing UAS-Ihog-YFP in either Hh-Gal4 (P compartment) or Ptc-Gal4 (A compartment) domains for 24-48h before dissection. The length extension of cytonemes was manually measured using the Straight tool from FIJI software. The statistical analysis and software simulations were done using a total of 984 cytonemes, 729 in the P compartment and 255 in the A compartment.

**Cell diameters for gradient normalization.**   Since the software computes the data in cell diameters, to compare the experimental data with model simulations, it is important to know the characteristic cell diameter in μm of each specific tissue studied. For the normalization of the gradient length, we manually measured approximately 100 cells along the X-axis in each wing imaginal disc (n = 19).

**Hh gradient imaging in wing discs.**   Hh protein gradient and the graded response of *ptc* enhancer reporter were measured experimentally using Plot profile tool of FIJI taken an

average Z-stack projection to get all the morphogen distribution along apicobasal sections of the wing disc epithelium. The fluorescence profiles of the corresponding channel for the *Hh*: *GFP BAC* and *EnhancerPtcRed* signals were measured in a 90x35 μm$^2$ region of the A compartment with the start positioned at ≈25 μm from the A/P border inside the P compartment.

**Mathematical protocol for the Hh gradient data.**   The Hh protein gradient and *ptc* enhancer reporter gradient response in wing discs and their experimental variabilities were estimated using 19 different wing disc samples, as follows: the background was estimated measuring the mean signal level over a 20 μm region in areas in which each reporter genetic tool is not active; for the *ptc* reporter signal, the region corresponds to the entire P compartment, while for Hh protein the region corresponds to the A compartment cells located away from the A/P compartment border. After subtracting the background, the intensity was normalized with the mean of the maximum intensities (3 values for the region of maximum *ptc* enhancer reporter expression and the whole P compartment signal for Hh protein levels). Finally, to compare the resulting data, we translated the measured profiles to the same reference origin; for the beginning of the Hh gradient we used the A/P compartment border. This origin was mathematically estimated using the well-defined sharp increase in *ptc* expression at the A/P compartment border. The graphic steps of the process are depicted in the S1 Fig.

## Data acquisition and quantification for *in vivo* imaging in abdominal histoblast nests

Pupal abdominal histoblasts imaging was performed in a chamber to seat and orient the pupae to look under the microscopy as described in [29]. The dorsal abdominal segment A2 was filmed using 40x magnification; Z-stacks of around 30 μm of thickness with a step size of 1.1–1.3 μm were taken using a LSM800 confocal microscope. The overnight movies of the Hh-GFP gradient and the *ptc-RFP* enhancer graded response during the histoblast migration (S3 Movie) were done recording Z-stacks every 2 min. For optimal recording of dynamic cytonemes, *in vivo* experiments using at the same time Gal4 and Q systems were taken in different conditions (Z-tack of 18.5 μm with a step size of 0.5 μm every-one minute). Also S2 Movie (see some sequences in Fig 3) was computationally treated using a deconvolution method (Huygens software) for cleaning the fluorescence signal.

**Cell diameters for gradient normalization.**   The cell diameters for Hh and *ptc* profile normalizations were measured for 14–32 different histoblasts per pupae along the X-axis (n = 9). Since we did not find statistical differences in cell diameters between A ($\phi^{anterior}$ = 4.285 ± 0.886 μm with n = 228) and P compartment cells ($\phi^{posterior}$ = 4.453 ± 0.887 μm with n = 245). We used the average value ($\phi$ = 4.37 ± 0.89 μm, n = 473) for abdominal histoblast simulations.

**Imaging Hh gradient in abdominal histoblast nests.**   The Hh protein gradient and *ptc* enhancer reporter signals were measured using the Plot profile tool of FIJI in an average Z-stack projection, as we have done for the wing imaginal disc samples. In each channel, profiles were measured for the same region of 200x130 pixels (51.51x33.48 μm$^2$) located in the A compartment close to the A/P compartment border.

**Mathematical protocol for Hh Gradient data.**   The experimental variability of Hh signal and its gradient in abdominal histoblast nests were estimated using 14 different regions (in the A compartment close to de A/P border) extracted from 9 pupae. In each sample, the background signal was subtracted using the corresponding minimum value and then the intensity was normalized with the maximum value in each case. Finally, the profiles were translated to the same position using the maximum as a reference.

**Statistical analysis of filopodia extensions.**   To study the parametric behavior of the data, we first performed a Shapiro-Wilk normality test. After testing the non-parametric condition

of the experimental distributions, we studied the significance using the Wilcoxon rank Sum test of homogeneity of variances (Implemented in Matlab2015a).

**Data analysis of cytoneme dynamics.** The experimental data of filopodia dynamics have been taken from previous studies [30]. Here, we have statistically studied in R language the differences between the times of the elongation, retraction and stationary phases of Triangular and Trapezoidal behaviors using a Kolmogorov-Smirnov statistical analysis (S2 Table), as we observed statistically significant differences between Triangular and Trapezoidal times we develop the model to consider both behaviors.

## Analysis of the Hh gradient formation by FRAP experiments

Fluorescence recovery after photobleaching (FRAP) is a method to study temporal evolution of fluorescent signals. An initial z-tack covering from apical to basal sections of the tissue was performed using Zeiss-LSM800 confocal microscopy to record the pre-breaching conditions of the sample. To avoid damaging the tissue, the photobleaching was done over a ROI of (48.7x61.3 μm$^2$) in a region located at the A/P compartment border (we used *ptc* expression as a reference for the A/P compartment border (Fig 7A)). Photobleaching of the abdominal histoblast nests was done by series of short exposions of 488 nm laser at 100% intensity until the signal at that z-plane reached less than 10% of the initial value. Since the Hh signal is present through apicobasal length of the tissue, we repeated the photobleaching conditions 7–10 times at different sections covering the total apico-basal tissue length. To obtain the Hh signal recovery over time we recorded the same region used in the pre-bleach z-stack conditions every 45 seconds immediately after photobleaching.

The acquired image samples of the photo-bleached ROI area were then treated with the imaging protocol described above (quantification of the wild type Hh gradient in abdominal histoblast nets). Since the resulting file is not a simple image but a temporal sequence of images, we automatized the process creating a macro script in Fiji that measures the Hh and Ptc profiles over time in the region where the photobleaching was performed. Since the experimental conditions can generate undesired photobleaching, a control of the Hh signal intensity in the P compartment was also measured each time. To study the FRAP recovery we used the previously described mathematical protocol that translates all signals to the same origin, but normalization was done using the P compartment control values as follow:

$$\frac{Intensity\ where\ FRAP\ was\ done - background\ signal}{control\ intensity - background\ signal}$$

This equation is used in FRAP experiments to mathematically remove the possible undesired photobleaching in the recovery measurements.

Finally, to visualize the recovery evolution of the pre-bleaching gradient in percentages, the resulting values were normalized to the pre-bleaching maximum value. All the samples studied (n = 6) showed the same recovery tendency for Hh and *ptc* expression profiles than the representative case shown in Fig 7B and 7C.

## Theoretical and computational methods

### Software code

Cytomorph was generated implementing the cytoneme model in Matlab language (MatlabR2015a). Since Cytoneme-mediated signaling has been reported for many different morphogens and for different animal systems, our goal was that Cytomorph could be used as a computational tool to help other scientist. We then decided to develop Cytomorph as an open

source software under a 3-clause BSD FOSS license. The software code and a manual for users are available in the software repository: https://github.com/AdrianA-T/cytomorph

The available version of Cytomorph has been divided into different modules that can be updated to incorporate new discoveries in the formation of gradients, these modules can also be remodeled to simulate specific requirements of the system under study.

## Units used in the model

The frame of reference selected for the model is summarized in Fig 1D. The distance expressed in terms of cell diameters was selected for two main reasons: first, it is an intuitive unit commonly used in biology, that helps to visualize the data; and second, for practical reasons, since the mathematical equation of the model and the software code implementation are simplified using this distance unit. Therefore, the distance estimated through the experimental data (initially in μm) was normalized to cell diameters dividing by the average cell size, as described in cell size measurement protocols. The temporal unit used in this model was the second, so, the time calculated in the model from the *in vivo* dynamic cytonemes were expressed in seconds. The rest of variables in the model are either dimensionless or expressed in terms of cell diameters and seconds.

## Requirements to compare experimental data of the gradient with the theoretical contact function

The conditions to compare *in silico* simulations with experimental data have been estimated in the S4 Text and can be summarized as:

- Mathematically: The degradation rate of the morphogen should be taken into account.

- Experimentally: Confocal images must have been taken according to the linear gamma function and within the limits of the acquisition range.

## *In silico* simulations

**Numerical simulations.** Each *in silico* prediction was computed 2000 times per simulation and case, the different values obtained over those 2000 times were used to generate the predicted gradient for those conditions. The standard deviation of the data obtained of those 2000 times was used as the expected signal variability. A more detailed study of the variability and fluctuations in Cytomorph can be found in S5 Text.

**Parameters and data for simulations.** S1 Table details the parameters used in each simulation. The updated experimental data in those simulations were obtained from the already described measurements of the wing disc cytoneme length and the average cell diameter ($\phi =$ 4.37 ± 0.89 μm for abdominal histoblasts and $\phi =$ 3.05 ± 0.65 μm for wing disc cells). The experimental cytoneme dynamics were obtained from previous studies [30]. Finally, the average Hh gradient profiles obtained in the validated simulations was used in both systems. The degradation rate of Hh was obtained from a previous study [66].

**Statistical analysis.** The Wilcoxon rank sum test was performed per pairs and the resulting p-values were graphically coded in matrixes with a green color that is graded depending on the significance. The code used was: black for no significance (= n.s) and dark green to light green respectively for the p-values: * = p-value < 0.05, ** = p-value < 0.01, *** = p-value < 0.001, **** = p-value < 0.0001.

As mentioned, *in silico* predictions were computed over 2000 times per simulation in each case; the different values obtained over those 2000 samples were used for the statistical analysis using Wilcoxon rank sum test (Implemented in Matlab2015a).

For the Coefficient of variation, we divided the 2000 simulations in 100 subgroups of 200 samples each. Then the coefficient of variation distribution per case was performed over those 100 subgroups. Finally, we used those 100 values per case for the posterior statistical analysis of the signal variability using the coefficient of variation.

**Simulations of the diffusion-degradation model.**   Simulations for the diffusion-degradation model (next equation) were performed using in Matlab2015a language using *pdepe* function for 1-D parabolic and elliptic PDEs (See S8 Fig).

$$\begin{cases} PDE : \dfrac{\partial u}{\partial t} = D\Delta u - \delta \cdot u \\[2mm] BC : u(0,t) = \langle u_{exp}^N \rangle_{x=0} \; 0 < t < \infty \\[2mm] \dfrac{\partial u(L,t)}{\partial x} = 0 \; 0 < t < \infty \\[2mm] IC : u(x,0) = 0 \; 0 < x \le L \end{cases}$$

The boundary and initial conditions are required to solve the equation. We decided to use the condition of the morphogen flux equal to zero at the tissue end (L) $\frac{\partial u(L,t)}{\partial x} = 0$. This derivate boundary condition is commonly used and in biological terms means that the morphogen cannot escape from the tissue. The initial conditions the morphogen gradient is zero $u(x,0) = 0$ since there is no previous diffusion of the morphogen. Finally, to have more precise experimental conditions for the diffusion simulations, the boundary condition at the origin has been selected as the experimental average of the maximum values for the normalized Hh gradient $u(0,t) = \langle u_{exp}^N \rangle_{x=0}$.

The same experimental data were used for both: cytoneme and diffusion models. A table detailing the parameters can be found in S3 Table.

## Cytomorph simulations for other morphogens

**Decapentaplegic (Dpp) gradient.**   The Dpp signaling pathway is functional in the same wing disc cells as Hh signaling. Therefore, we used the length distribution of cytonemes protruding from the A and P compartment cells quantified in this work using non-morphogen-specific markers (membrane markers) (Fig 3D). Since some of the receiving cells are also producing Dpp in the A compartment of the wing disc, we only simulated the gradient in the Dpp receiving cells of the P compartment.

Although we have not focused on the temporal aspects of the Dpp gradient formation, for the proper simulation of the Dpp gradient we considered the experimental degradation rate of Dpp [52]. It has been reported that the experimental half-time of the Dpp protein was shorter than that of the Hh protein (Dpp = 45 min [52] vs Hh = 166 min [66]), therefore, we simulated the Dpp gradient during a short period of development. The exact values of the parameters simulated are available in S1 Table. To be able to compare our simulations with the experimental Dpp gradient, we used the WebPlotDigitizer tool to convert images in numerical data extracting the published Dpp experimental information from [50].

**Wingless (Wg) gradient extension.**   The *in silico* simulation of the exact shape of the Wg gradient in the wing disc would require to update Cytomorph with new modules because some of the receiving cells are also producing Wg. We calculated the expected length of the Wg gradient using the spatial conditions of Eq-3, in which the range of the gradient is limited by the sum of the lengths of cytonemes emanating from producing and receiving cells: $x_p < \lambda_r(t) + \lambda_p(t) - x_r$. Therefore, the total length of the gradient is limited by maximum length of cytonemes $\lambda_{max,r} + \lambda_{max,p}$ protruding from the border of producing/non-producing cell.

Assuming again that the receiving and producing lengths of Wg cytonemes are the same in the wing disc as those of others morphogens ($\lambda_{mean,r}$ = 11.8 $\mu m$ with a standard deviation of 4.1 $\mu m$ and $\lambda_{mean,p}$ = 14.5 $\mu m$ a standard deviation of 5.5 $\mu m$), the expected length of the Wg gradient will be $\lambda_{max,r}+\lambda_{max,p} \approx$ 15.9+20 = 35.9 $\mu m$ from the last Wg producing cell, which corresponds to ~42 $\mu m$ from the D/V border.

## Supporting information

**S1 Text. The dynamics of cytoneme lengths.**
(DOCX)

**S2 Text. The probability of contacts.**
(DOCX)

**S3 Text. Cytomorph software architecture.**
(DOCX)

**S4 Text. Comparing experimental and *in silico* morphogen gradients.**
(DOCX)

**S5 Text. Variability and fluctuations in Cytomorph.**
(DOCX)

**S1 Fig. Graphical representation of mathematical treatment for Hh and Ptc experimental expression profiles.** The normalization processes for Ptc (top) and Hh (bottom) profiles described in Materials and Methods are represented. The first column show the raw data obtained using the FIJI plot profile tool. The second column illustrates the normalized signal in cell diameters without background. The third column represents the same normalized data translated to the common origin at the A/P compartment border, using the beginning of Ptc expression to locate it. The final column shows the average (continuous thin lines) and the variability of the experimental samples (standard deviations are plotted in color shaded areas, Ptc in red and Hh in green).
(PDF)

**S2 Fig. The shape of the gradient is a consequence of the contact distributions.** The grafts show that the spatial distribution of contacts in the receiving cells (A) together with its temporal evolution (B) determine the final shape of the gradient (C).
(PDF)

**S3 Fig. Influence of the number of producing cell rows in the shape of the gradient in abdominal histoblast nests.** The grafts show how different number of producing cell rows change the gradient properties: amount of the transmitted morphogen (A), signal variability (A') and scaling (A"). Since the length of the cytonemes and the size of the Hh producing cells in abdominal histoblast nets are smaller than those in imaginal wing discs, these results confirm the behavior observed in imaginal wing discs but with a scaling factor.
(PDF)

**S4 Fig. Influence of the number of receiving cell rows in the shape of the gradient.** These results show that increasing the number of A compartment cell rows does not change the gradient properties: amount of transmitted morphogen (A), signal variability (A') and scaling (A"). These results are not surprising since the amount of Hh taken by each receiving cell of the A compartment only depends on the contacts of that cell with the producing cells of the P compartment.
(PDF)

**S5 Fig. Influence of the coordination of cytonemes growth and contact dynamics in gradient formation. (**A) Simulations showed that dynamic and static cytonemes generate different gradient shapes. The differences suggest that cytoneme dynamics play a pivotal role in shaping the Hh gradient (A"). Simulations for static and dynamic cytonemes (dynamic with "tc = 60s" in red, static with "tc = 120s" in light blue, static with "tc = 300s" in dark blue) showed that the final amount of morphogen released depends on factors, such as the time taken for a contact to be effective (time-length contacts "tc"). (B) Growth simulations of triangular (t) and Trapezoidal (T) dynamics of cytonemes contacting while growing (= 1) or contacting at maximum elongation (= 0) (Ref.case T = t = 1 in red, case 2: T = 0,t = 1 in light blue, case 3: T = 1,t = 0 in blue, case 4: T = 0,t = 0 in dark blue). The simulations shown for contact during growth using each type of behavior, triangular or trapezoidal, result in similar effects without statistically significant differences (C2 and C3 in B, B 'and B' '). However, if both types of cytoneme behaviors happen together, either contacting while growing or just contacting after growth (C-R and C4 respectively), there are relevant differences in the amount of morphogen transferred (B), the signal variability (B') and the gradient shape (B").
(PDF)

**S6 Fig. Signaling compensation between the number of producing cells and the number of cytonemes per cell.** Experimental data are represented in green, reference simulation for a normal tissue ($N_P$ = 15 and $n_{cyt}$ = 4) in red, altered tissue with less producing cells ($N_P$ = 3 and $n_{cyt}$ = 4) in light blue and altered tissue with a compensatory mechanism for this cell number reduction ($N_P$ = 3 and $n_{cyt}$ = 7) in dark blue. (A) Left, morphogen distribution (y-axis) along the receiving cells normalized to the maximum value of the reference case. Right, violin plots for the number of contacts in the first row of receiving cell $x_0$, normalized to the reference average value (2000 simulations per case). A') Violin plots for the coefficient of variation (y-axis) per case (x-axis) in the first row of receiving cells ($x_0$). A") Left, distribution of contacts normalized to their maximum (y-axis) to compare the changes in the shape in the receiving cells. Right, coefficient of the previous normalized distributions of contacts (y-axis) to study the scaling in the receiving cells.
(PDF)

**S7 Fig. Cytomorph simulations for Dpp gradient in imaginal wing disc of Drosophila. (**A) Cytomorph simulations for Dpp gradient (blue) compared to the real experimental Dpp gradient [50] (average in yellow and experimental variability in grey). (B) Dpp cytonemes contacting while growing (CWG, blue) or contacting after growth (CAG, red). The simulations show that two types of dynamics predict relevant differences in the amount of morphogen transferred (B), the signal variability (B') and the gradient shape (B"). Our results suggest that cytonemes contacting after growth fits better the experimental Dpp gradient.
(PDF)

**S8 Fig. Matlab resolution of diffusion-degradation equation.** Screenshot of the computational code used for the simulation of the diffusion-degradation model. Left) Main script with *pdepe* function for 1-D parabolic and elliptic PDEs. Right) Three auxiliary functions called by *pdepe* that contain: the diffusion-degradation equation (top), the boundary conditions (middle) and the initial conditions (bottom) as described in Material and Methods.
(PDF)

**S1 Movie. Cell organization at the A/P compartment border in abdominal histoblast nests.** It shows a confocal Z-stack from apical to basal of an A2 abdominal histoblast nest of a *Drosophila* pupa, with the A compartment marked with life-actin-RFP (Red) and the P

compartment marked with CD8GFP (Green). Scale bar: 10 μm.
(AVI)

**S2 Movie. *In vivo* movie of wild type abdominal histoblast cytonemes.** *In vivo* abdominal histoblast cytonemes located at the basal side of the tissue; confocal sections were taken at one-minute interval. The left movie shows merge channels of A compartment cells labeled with life-actin-RFP (in red) and of P compartment cells are marked with CD8GFP (in green). The movie located in the middle shows a single channel corresponding to A compartment cytonemes, and the right movie shows a single channel corresponding to P compartment cytonemes. Scale bar: 15 μm.
(AVI)

**S3 Movie. Abdominal histoblast nest proliferation and migration.** While migration and proliferation of dorsal A and P compartment histoblast nests take place, larval epidermal cells are eliminated. The protein marked *Hh*:*GFP BAC* allows the visualization of Hh in the producing cells (P compartment histoblast nest, in green). The *EnhancerPtcRed* reporter correspond to *ptc* transcriptional activity in response of Hh gradient; this allows the visualization of the Hh gradient response in the receiving cells (A compartment histoblast nests, in red). Scale bar: 30 μm.
(AVI)

**S1 Table. Parameters used in cytoneme model simulations.** Numerical parameters used in the cytoneme model for each simulation. When a scan of a single variable is performed, the values are written from the initial to the final values with a specific step between them. Binary values mean that this condition is on = 1 or off = 0. WD and AH are abbreviations for wing imaginal discs and abdominal histoblast nests and T and t for Trapezoidal and Triangular behaviors respectively. Units are specified at the top of each column.
(PDF)

**S2 Table. Statistical study of the phase times used to compare cytoneme dynamics.** Statistical p-values used to compare elongation, retraction and stationary phases using a Kolmogorov-Smirnov statistical analysis (n.s = no significance).
(PDF)

**S3 Table. Diffusion parameters used in each simulation.** Numerical parameters used in the Fig 4 for diffusion simulations. Units are specified at the top of each column.
(PDF)

## Acknowledgments

We are grateful to Pedro Ripoll, Ana-Citlali Gradilla, Nicole Gorfinkiel, David G. Míguez, and Juan Soler for their advice and comments on the manuscript. Thanks to Dagmar Iber for hosting A.A-T and for her advice. We also thanks to Confocal Facilities of the CBMSO and to Bloomington and Vienna stock centers for fly stocks.

## Author Contributions

**Conceptualization:** Adrián Aguirre-Tamaral, Isabel Guerrero.

**Data curation:** Adrián Aguirre-Tamaral.

**Formal analysis:** Adrián Aguirre-Tamaral.

**Funding acquisition:** Isabel Guerrero.

**Investigation:** Adrián Aguirre-Tamaral.

**Methodology:** Adrián Aguirre-Tamaral.

**Project administration:** Isabel Guerrero.

**Software:** Adrián Aguirre-Tamaral.

**Supervision:** Isabel Guerrero.

**Validation:** Adrián Aguirre-Tamaral.

**Visualization:** Adrián Aguirre-Tamaral.

**Writing – original draft:** Adrián Aguirre-Tamaral.

**Writing – review & editing:** Adrián Aguirre-Tamaral, Isabel Guerrero.

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
