## [Decision Letter · Decision Letter 0]

18 Jun 2021

Dear Prof. Guerrero,

Thank you very much for submitting your manuscript "Improving the understanding of cytoneme-mediated morphogen gradients by in silico modeling" for consideration at PLOS Computational Biology. As with all papers reviewed by the journal, your manuscript was reviewed by members of the editorial board and by several independent reviewers. The reviewers appreciated the attention to an important topic. Based on the reviews, we are likely to accept this manuscript for publication, providing that you modify the manuscript according to the review recommendations.

Sincerely,

Jason M. Haugh

Deputy Editor

PLOS Computational Biology

[LINK]

Reviewer's Responses to Questions

**Comments to the Authors:**

Reviewer #1: In this version of the manuscript the authors have sufficiently addressed most of the concerns I raised during the first round of reviews. However, there are still important details about the physical mechanisms being modeled that remain absent. Most importantly, the dynamics of how either producing or receiving cells create new cytonemes is left critically underdescribed. The GUI interface has a parameter labelled "Number of cytonemes per cell", which I presume is the number of cytonemes each cell is intended to grow, but this does not appear to be strictly stated. Additionally, after a cytoneme completes its life cycle of growing and retracting and possibly pausing in between is it immediately replaced by a new cytoneme that undergoes the same dynamics? If not, is there some form of time lag between one cytoneme fully retracting and another beginning its extension from the same cell and what determines this time lag? Given that an explicit understanding of the dynamics of the birth and death of cytonemes is crucial for the existence of a well defined steady state morphogen distribution profile, it should be clearly outlined in the manuscript.

A further question that is immediately raised once the birth-death dynamics of the cytonemes within the model is established is how do these dynamics compare to those observed in the experiments. What are the statistics of how many cytonemes each cell has active at any given time or has such a distribution not been sufficiently measured? A discussion as to how frequently cytonemes are created and removed from the system and its effect on the number of cytonemes each cell currently has extended would not only provide a useful understanding of how accurately the model used in Cytomorph is replicating this distribution but could also potentially introduce a path towards more precise measurements being made in the future to better understand and more precisely simulate these dynamics.

Finally, the authors should make sure to carefully spell check the manuscript. I noticed several places in the Discussion section where pronouns appear to have been dropped and the word "established" was misspelled as "stablished".

Reviewer #2: I commend the authors for their work and have only one request: the data showing that cytonemes move morphogens between cells and are responsible for morphogen exchange between producing and receiving cells is very strong. This is not therefore "a model" and should not be described as a "proposal".

Reviewer #3: The manuscript has strongly improved, all my suggestions and comments have been addressed. Therefore, I support publication of the manuscript.

**Have the authors made all data and (if applicable) computational code underlying the findings in their manuscript fully available?**

Reviewer #1: Yes

Reviewer #2: Yes

Reviewer #3: Yes

PLOS authors have the option to publish the peer review history of their article (what does this mean?). If published, this will include your full peer review and any attached files.

Reviewer #1: No

Reviewer #2: No

Reviewer #3: No

Figure Files:

Data Requirements:

Reproducibility:

References:

---

## [Editor Report · Decision Letter 1]

3 Jul 2021

Dear Prof. Guerrero,

We are pleased to inform you that your manuscript 'Improving the understanding of cytoneme-mediated morphogen gradients by in silico modeling' has been provisionally accepted for publication in PLOS Computational Biology.

Best regards,

Jason M. Haugh

Deputy Editor

PLOS Computational Biology

---

## [Editor Report · Acceptance letter]

29 Jul 2021

PCOMPBIOL-D-21-00938R1 

Improving the understanding of cytoneme-mediated morphogen gradients by in silico modeling

Dear Dr Guerrero,

I am pleased to inform you that your manuscript has been formally accepted for publication in PLOS Computational Biology. Your manuscript is now with our production department and you will be notified of the publication date in due course.

With kind regards,

Agota Szep
